# Charting the N-Terminal Acetylome: A Comprehensive Map of Human NatA Substrates

**DOI:** 10.3390/ijms221910692

**Published:** 2021-10-02

**Authors:** Petra Van Damme

**Affiliations:** iRIP Unit, Laboratory of Microbiology, Department of Biochemistry and Microbiology, Ghent University, K. L. Ledeganckstraat 35, 9000 Ghent, Belgium; petra.vandamme@ugent.be; Tel.: +32-9-264-51-29

**Keywords:** N-alpha acetyltransferase (NAA), N-terminal acetylation (NTA), N-terminal proteomics, NAA10, NAA15, NAA50, NatA, NatE, ribosome

## Abstract

N-terminal acetylation (Nt-acetylation) catalyzed by conserved N-terminal acetyltransferases or NATs embodies a modification with one of the highest stoichiometries reported for eukaryotic protein modifications to date. Comprising the catalytic N-alpha acetyltransferase (NAA) subunit NAA10 plus the ribosome anchoring regulatory subunit NAA15, NatA represents the major acetyltransferase complex with up to 50% of all mammalian proteins representing potential substrates. Largely in consequence of the essential nature of NatA and its high enzymatic activity, its experimentally confirmed mammalian substrate repertoire remained poorly charted. In this study, human *NatA* knockdown conditions achieving near complete depletion of *NAA10* and *NAA15* expression resulted in lowered Nt-acetylation of over 25% out of all putative NatA targets identified, representing an up to 10-fold increase in the reported number of substrate N-termini affected upon human NatA perturbation. Besides pointing to less efficient NatA substrates being prime targets, several putative NatE substrates were shown to be affected upon human NatA knockdown. Intriguingly, next to a lowered expression of ribosomal proteins and proteins constituting the eukaryotic 48S preinitiation complex, steady-state levels of protein N-termini additionally point to NatA Nt-acetylation deficiency directly impacting protein stability of knockdown affected targets.

## 1. Introduction

Co- and post-translational protein N-terminal acetylation (Nt-acetylation or NTA) exemplifies an omnipresent and abundant protein modification with one of the highest stoichiometries in mammalian cells reported for protein modifications to date [1,2]. Nonetheless, NTA contributes to an increase in mammalian N-terminal proteoform diversity by the occurrence of partially NTA protein N-termini [3]. NTA is carried out by a family of N-terminal acetyltransferases or NATs which belong to a subfamily of GNATs (GCN5 related N-acetyltransferase fold), with GNATs representing one of the largest protein superfamilies with over 10,000 members reported. To accomplish NTA, the acetyl moiety of acetyl-CoA (Ac-CoA) is being transferred to a protein’s free N-terminus by the action of the NAT.

To date, 12 NATs—including 8 distinct eukaryotic NATs NatA to NatH next to 1 archaeal and 3 bacterial NATs—with evolutionary conserved folds and related catalytic mechanisms have been identified [4,5]. These NATs vary greatly in their (subunit) composition and substrate specificities. For the vast majority of identified NATs, structural and biochemical characterizations are at hand, thereby providing molecular insights into their mechanisms of action and modes of regulation besides revealing molecular determinants steering the by and large unique substrate-specific activity potential of NATs (reviewed in [4] and reported recently for NatC [6]). While only a partial redundancy in the specificity profiles of certain NATs could be observed, the targeted protein N-termini per NAT appear by and large unique, with only some exceptions as recently proven for the yeast Lge1 protein representing the first redundant yeast NatA/NatD substrate identified to date [7]

While NTA has been considered a possible bystander modification for certain modified targets, by the changed chemical properties of NTA protein N-termini (e.g., charge neutralization, increased hydrophobicity and size), diverse biological outcomes comprising alterations in protein complex formation, folding, localization and stability have also been attributed to protein NTA [8]. Ascribed to the vast number of substrates being targeted by most NATs, complex distinct phenotypes have been observed in case of eukaryotic NAT knockouts and NAT activity has also been implicated in human diseases, cancer, development and plant stress responses [9,10,11].

With the NatA complex accounting for the NTA of up to 40 to 50% of all mammalian proteins, NatA represents the major NAT [1]. Moreover, NatA is unique in that it is the sole co-translational acting eukaryotic NAT besides the very substrate selective NAT NatD, which relies on the prior removal of the initiator methionine (iMet) by the action of methionine aminopeptidases (MetAPs). iMet removal, a process referred to as N-terminal methionine excision (NME), represents another nascent protein modification which takes place at the ribosome during the initial stages of protein biogenesis [12,13]. A kinetic competition between NATs and MetAPs, and thus mutual influencing aspect of NME and NTA, has previously been demonstrated [14]. More specifically, NatA targets iMet-processed N-termini harboring a relatively small and uncharged residue (i.e., Ser-, Ala-, Gly-, Thr-, Val- and Cys-) at position P1’ [2,15].

NatA is a heterodimer consisting of the from yeast to human evolutionary conserved catalytic N-alpha acetyltransferase (NAA) NAA10 and the large ribosome anchoring auxiliary subunit NAA15 [2,16,17,18]. Besides, an involvement of ribosomal RNA expansion segments for anchoring NatA to the ribosome and positioning it dynamically below the nascent polypeptide exit tunnel has recently been demonstrated [19]. 

These ribosomal associations, together with the finding that opposed to wild type NAA15 a ribosome-binding mutant of NAA15 was unable to rescue the temperature-sensitive growth phenotype in yeast deleted for NatA [20], and the observed NAA10 dependent stabilization of NAA15 [21], all point to the co-translational dependent aspect of NatA-mediated NTA in vivo. Further, a role for inositol hexaphosphate (IP6) in human NatA stability and activity has been demonstrated [22,23].

Intriguingly, monomeric NAA10 is incapable of acetylating canonical NatA substrates and displays an altered substrate specificity [21,24]. More precisely, at least in vitro, monomeric NAA10 exhibits specificity for acidic N-terminal sequences [24], a finding supported by the different catalytic strategy suggested for free NAA10 [21].

Another NAA catalytic subunit—namely, NAA50, also complexes with NatA via hydrophobic interactions with NAA15. More specifically, in HeLa cells, 20% of endogenous NAA50 was shown to be associated with NatA [25]. This NAA10/NAA15/NAA50 complex is collectively referred to as NatE. Curiously, while the archael *Sulfolobus solfatarics* NAT (*Ss*NAT) was put forward as an eukaryotic NAA10/NAA50 hybrid [26], NAA10 and NAA50 were shown to mutually influence each other’s activity when residing within NatE [22,27,28]. Further, in contrast to monomeric NAA15, monomeric NAA10 and human and *Drosophila* NAA50 are stable and can operate independent of NatA. In this context, a wide panel of moonlighting catalytic and non-catalytic functions have been ascribed to non-NatA complexed NAA10 [14,29].

Besides NAA50, the chaperone protein Huntingtin-Interacting Protein K (HYPK) signifies another stable NatA/NatE interactor (NAA10/NAA15/NAA50/HYPK or NatE/HYPK) even though a mutual allosteric hindrance for binding of HYPK and NAA50 was reported [22]. In contrast to the demonstrated requirement of HYPK for proper NTA of the known in vivo NatA substrate protein PCNP [30], recent reports demonstrate that HYPK displays NatA inhibitory activity [22,23]. All considering, a regulatory role of HYPK over the NatA and NatE complex to ensure rigorous quality control over NatA/NatE-mediated NTA has been proposed.

In recent years, a wealth of data on *NAA10* and *NAA15* mutations associated with various human diseases characterized by severe phenotypes has accumulated. One well studied mutation representative of additional pathogenic *NAA10* variants is the Ser37Pro (S37P) mutation in the gene encoding NAA10 (i.e., *NAA10* S37P). *NAA10* S37P is characterized by reduced NatA catalytic activity and NTA due to impaired NatA complex formation [31,32,33]. Presenting symptoms of this and additional pathogenic *NAA10* variants have now collectively been referred to as *NAA10*-related syndrome [34].

While non-essential in *S. cerevisiae*, NAA10 catalysed NTA has been reported as essential in a wide variety of higher eukaryotic species (e.g., *Caenorhabditis elegans*, *Drosophila melanogaster*, *Danio rerio* and *Arabidopsis thaliana*). Notwithstanding, strong effects such as slow growth, mating and decreased survival have been observed under stress conditions in yeast deletion strains [9]. In consideration it was unexpected that *Naa10* KO mice were viable. Since functional redundancy was previously demonstrated for the NAA10 homolog NAA11 in mice and humans, intuitively NAA11 was hypothesized to account for the loss of *Naa10* whilst not in accordance with its restricted expression profile observed [35]. A recent study by Kweon et al. however univocally demonstrates a compensatory role of the newly discovered and ubiquitously expressed paralogue *Naa12* in mice in vivo, as in contrast to *Naa10* KO mice, a *Naa10 Naa12* double KO in mice turned out to be embryonically lethal [36].

By making use of CRISPR/Cas9 gene editing, *NAA15* haplo-insufficient and deficient induced human pluripotent stem cells (iPSCs) could successfully be obtained. Interestingly, in these iPSCs, a mere partial effect on NatA-dependent NTA acetylation was observed [37]. In case of *NAA15* deficiency and since a role for NAA16-NAA10 complexes has been proposed to act as a potential backup system for the NAA15-NAA10 NatA complex [38], akin NAA11 expression, expression of its homologue NAA16 in human cells may also (in part) be responsible for this observation.

Potential redundancy and high NTA efficiency overall have complicated studies aiming at capturing the NatA substrate repertoire in a comprehensive manner further enabling distinguishing efficient from less efficient NAT substrate N-termini.

In the current study, we report on improved human NatA knockdown conditions achieving a significant depletion of NAA10 and NAA15 expression. Using N-terminal proteomics, lowered NTA of at least 25% out of all putative NatA targets identified could be observed. Besides, several previously reported and putative novel NatE substrates were found and a lowered expression of ribosomal proteins and proteins implicated in translation initiation was observed. Finally, a significant impact on protein stability of knockdown affected NatA targets was noticed.

In sum, our study points to the need of a highly significant lowered functional expression of human NatA to comprehensively probe its Nt acetylome and its proteome-wide impact.

## 2. Results

### 2.1. N-Terminal Proteomics to Explore the Human Nt-Acetylome upon Pronounced NatA Knockdown

We characterized the in-vivo substrate specificity of the human NatA complex by comparing the Nt-acetylomes of A-431 cells treated with si*NAA10*/si*NAA15* pool (si*NatA*) versus siControl (si*CTR*) treated cells (Figure 1A) by means of N-terminal proteomics. By preforming two consecutive (reverse) siRNA transfections in the presence of pan-caspase inhibition, a very efficient knockdown of more than 99% could be attained, as evidenced by the near absence of NAA10 and NAA15 expressed (Figure 1B and Appendix A). Since induction of apoptosis was previously reported in human NatA knockdown studies [39], apoptotic cell death was specifically blocked by Z-VAD-fmk.

In line with previous reports [21,27,37], and as evidenced from protein expression levels when probing for the NatA subunits by means of Western blotting, exclusive knockdown of *NAA10* also results in a significant destabilization of NAA15 (Appendix A).

When performing N-terminal proteomics in combination with differential L-Arg stable isotope-labeling by amino acids in cell culture (SILAC) [40] and in-vitro ^13^C_2_D3-acetylation [41,42], the extent of NTA and relative quantification of N-terminal peptides can be determined [43,44]. In the setup analyzed, the si*NatA* and si*CTR* samples were labeled with a light (^12^C_6_-L-Arg) and heavy isotopic variant of L-Arg (^13^C_6_^15^N_4_-L-Arg), respectively (Figure 1).

Following LC-MS/MS analysis, we identified 1498 unique N-termini originating from 1422 human proteins that were in compliance with the rules of NTA and NME [45,46], thereby representing proxies of translation initiation events (Appendix A and Materials and Methods). Of these, 1482 N-termini started at position 1 or 2, thus matching database annotated N-termini, whereas 16 started beyond position 2, the latter indicative of database non-annotated N-termini generated upon alternative translation initiation or translation of alternative (unknown) transcripts [47].

Of the 1306 identified N-termini carrying this modification (as inferred from MS or MS^2^ data (Appendix A)) and representing 87% of all identified N-termini (a percentage in line with previous reports [2,32]), 182 N-termini displayed a reduction in the degree of NTA upon h*NatA* knockdown (Figure 2 and Appendix A).

Expectedly, of these, the grant majority—173 or 95%—of affected N-termini displayed the NatA substrate specificity (Figure 3, Table 1 and Table 2), meaning that more than 25% out of the putative NatA targets in the si*CTR* setup (i.e., 173 of 687 NTA NatA type N-termini) were susceptible to si*NatA* knockdown (Table 1 and Table 2). Notably, in our original study only 7% of all putative hNatA substrates were affected by the reduced level of hNatA (i.e., 16 of 242) and this despite a significant, but somewhat lower, *NatA* knockdown efficiency [2]. Overall, this demonstrates the need for excessive NatA depletion or NatA perturbation to comprehensively probe for human NatA substrates in a cellular context.

MS spectra of representative si*NatA* affected NatA Ala- substrate N-termini are shown in Figure 4 and Appendix A and among others, include the translational activator GCN1 (^2^AADTQVSETLKR^13^) as a representative and previously confirmed NatA substrate [2,32,37], besides two newly identified mitochondrial protein substrates; namely, alpha-ketoglutarate-dependent dioxygenase alkB homolog 7 and import inner membrane translocase subunit Tim8 B (Figure 4, panels A, B and D). 74% of all si*CTR* NTA NatA type N-termini (e.g., ^2^ASELEPEVQAIDR^14^ of the E3 ubiquitin-protein ligase UBR2 (Figure 4C)) however remained unaffected (Table 1).

Besides affected NatA type N-termini, 9 non-NatA type N-termini displayed a reduced degree of NTA (Figure 2, Appendix A). More specifically, the latter category of affected N-termini belongs either to iMet-retaining NatA type N-termini (7 N-termini); five MV- and two MT- N-termini (NAT category “other”), besides two ML- N-termini (NAT category “NatC”). Moreover, this set includes two N-termini of which the degree of NTA was previously found to be reduced in patient derived fibroblasts and/or B-cells harboring the Ogden syndrome mutation in the gene encoding *NAA10* (i.e., *NAA10* S37P) [32] (Appendix A). Since besides a reduced catalytic capacity of NatA and the decreased NTA of a subset of NatA and NatE substrates, an impaired interaction between human NAA10 S37P and NAA50 (NatE) was demonstrated in Ogden cells, our data hints to these 9 non NatA N-termini as being highly likely NatE substrates (Figure 2, Appendix A).

In contrast to the general lower NTA levels of previously reported NatA substrates here identified in our si*NatA* setup [2,32,37], it is noteworthy that presumed NatE substrates were in general (somewhat) less affected in their NTA levels (e.g., in the case of p53 and DNA damage-regulated protein 1 (ML-) and Peptidyl-prolyl cis-trans isomerase A (MV-)), or even unaffected (when determined) in case of Kinesin-like protein KIF21A (ML-), ARL14 effector protein (MM-) and SUMO-activating enzyme subunit 1 (MV-). These results signify that the impaired interaction between NAA50 and NatA in the context of Ogden patients likely has a more prominent negative impact on NatE NTA as compared to the destabilization of NAA50—and thus NatE activity—in the *NatA* knockdown conditions employed in this study.

### 2.2. hNatA Affects Translation, Lipid Metabolism and Regulation of Cyclin-Dependent Protein Kinase Activity

When performing a comparative statistical analysis of N-terminal peptide MS-signal intensities in the si*CTR* versus si*NatA* setup (see Materials and Methods), 76 out of 1415 unique N-termini with a determined expression ratio displayed significantly altered steady state levels (*p* ≤ 0.01). Among the regulated N-termini, the Nt-free N-terminal peptide of NAA15 (^2^PAVSLPPKENALFKR^16^) was found as the N-terminus of which the expression which was most significantly reduced (over 7-fold) upon si*NatA* knockdown (Appendix A). More specifically, the levels of 27 and 49 protein N-termini were respectively lowered and elevated upon si*NatA* knockdown (Appendix A). Interestingly, among the regulated proteins, and out of the 24 regulated NatA type N-termini with determined degrees of NTA (29 regulated NatA N-termini in total), 11 N-termini (46%) were here identified as NatA substrate N-termini. Overall, this indicates a ~2-fold enrichment of NatA substrates in the category of regulated over non-regulated NatA type N-termini (i.e., 171 NatA substrate N-termini in case of 716 non-regulated NatA N-termini (24%) with determined degrees of NTA). This observation was statistically significant as determined by a chi-square test of independency (*p* < 0.02), indicating that deficiency of NatA NTA may thus directly impact protein stability and concomitantly the steady-state expression levels of certain target protein.

Out of the 11 regulated NatA substrate N-termini, 3 and 8 respectively displayed a significantly decreased or increased levels upon NatA depletion. A representative example of a NatA substrate N-terminus originating from the mitochondrial alpha-ketoglutarate-dependent dioxygenase alkB homolog 7 (^2^AGTGLLALR^10^), classified as being fully NTA in the control setup while being only partially NTA (80%) in the si*NatA* knockdown sample, was found significantly upregulated (*p* ≤ 0.01) upon si*NatA* knockdown (ratio 2.6) (Figure 4, panel D).

As inferred from a ranked GO gene enrichment analysis using GOrilla [50] and considering the highest ranking instance per gene, in line with a recent proteomics study performed in *NAA15* haplo-insufficient and deficient induced pluripotent stem cells (iPSCs) [37], a general and significant downregulation of ribosomal proteins clustering near the ribosomal NatA docking site and head of the small ribosomal subunit could be observed besides a lowered expression of proteins constituting the eukaryotic 48S preinitiation complex (Figure 5 and Appendix A). Overall, these altered N-terminal steady-state levels appear to affect translation (initiation). Moreover, cytoskeleton protein and integral membrane components next to lipid metabolic and cyclin-dependent protein kinase activity regulation processes were significantly upregulated in the si*NatA* setup (Appendix A).

## 3. Discussion

In line with previous reports [1,2,32] 87% of the proteome was here shown to carry an acetyl moiety at their N-terminus.

While in our original study merely 7% hNatA substrates were found to be affected by the reduced level of hNatA despite a significant knockdown efficiency (i.e., 30% and 5% of normal NAA15 and NAA10 level, respectively) [2], in the current study, pronounced NatA knockdown—as evidenced by the nearly undetectable levels of NAA10 and NAA15—pointed to 25% out of the putative NatA targets with lowered NTA.

More so, the number of human NatA substrate N-termini identified in our study vastly exceeds the numbers reported in related N-terminomics efforts aiming at identifying human NatA targets as no less than 182 unique N-termini were here proven affected. This number thus greatly contrasts the mere 16 and 34 affected N-termini reported in a previous NatA knockdown study in Hela cells [2] and in null and/or haplo-sufficient NAA15 iPSC cells, respectively. The latter number in iPSCs could however be increased to 42 when considering the NTA rules applied here and in [2,43] to assign NAT substrates [37]. All in all, in the current effort to chart for NatA targets in human cells, an over 4 to 11-fold increase in the number of affected N-termini by NatA could be identified.

This increase in NAT targets identified are in line with some of our previous study results reporting on affected substrate N-termini in HeLa si*NatB* knockdown proteomes. Here, a more efficient knockdown resulted in an over 4-fold increase of NatB-type substrates displaying reduced NTA (i.e., 9% vs. 39% of the potential NatB-type substrates displayed a reduced Nt-acetylation upon pronounced knockdown), indicating a strong correlation between NAT and NTA levels [2,37,43,44]. Moreover, these reports firmly strengthen that suboptimal NAT substrates are primarily affected upon NAT perturbation.

That levels of NatA can be lowered drastically before any NTA effect becomes apparent is quite striking and might possibly be explained by the low levels of ribosome-associated and complexed NatA required to efficiently carry out NTA. Moreover, associations between NAA15/NAA10/NAA50 (NatA/NatE) and even associated HYPK (NatE/HYPK) have been shown to have a mutually (functionally) stabilizing effect on top of influencing the catalytic activities of both NAA10 and NAA50 ([14,21,27,28,30,37] and this study). These stabilizing effects mean that complexed NatA/NatE is more efficient and stable, and thus potentially lowered less drastically in concentration upon knockdown as compared to the concentration of its composing subunits present in monomeric form at early timepoints post knockdown.

Interestingly, while 95% affected N-termini displayed the NatA substrate specificity, 7 iMet-retaining NatA type N-termini were affected. This substrate specificity is indicative of them being putative NatE substrates likely affected by an impaired interaction between NatA and NAA50 (NatE) in case of NatA perturbation [32]. Besides, coordinated (and thus influenced) NTA actions by both NatE catalytic subunits have been demonstrated [22].

The observation that abolishing NAA15 expression in *NAA15* null iPSCs affects the NatA Nt-acetylome less pronounced than the NatA knockdown model assessed in this study might indicate a compensatory role for the NAA15 paralogue NAA16 in human cells when considering cotranslational NatA Nt-acetylation [38]. While nonetheless NAA16 expression was low in iPSCs and generally less abundant as compared to abundant *NAA15* [38], our previous and current knockdown studies demonstrates that limited levels of NAA10 and NAA15—and thus possibly also by extension low NAA16 levels—might still be sufficient to maintain the essentially unaltered NTA levels of NatA substrates reported for *NAA15* null iPSCs.

However, it is noteworthy that all 3 NatE constituents are listed “core essentialome” components in case of the human haploid HAP1 and the near-haploid KBM7 cell lines [51]. In accordance, we were unable to successfully obtain single *NAA15*, *NAA10* or *NAA50* haploid HAP1 CRISPR knockout cell lines (unpublished data), indicating a cell-type dependent rather than a general compensatory role for NAA16 correlating with *NAA16* expression levels [38].

Alternatively, in *NAA15* null cells, NAA10 could also carry out post-translational Nt-acetylation independent of NAA15. This scenario is however very unlikely for at least 3 reasons. First, a different substrate specificity profile of non-complexed NAA10 was reported [21,24]. Second, a clear overlap of the substrate repertoire identified in the NAA15 iPSCs [37] and a previous as well as current NatA knockdown study was notable, and third, ribosome interaction has been proven a crucial aspect for NatA activity in yeast [20].

Remarkably, in contrast to previously reported proteome-wide findings, our results indicate that deficiency of NatA NTA may directly impact protein stability and thus the expression levels of NatA targets. Especially, the presumed less efficient or suboptimal NatA—and thus NatA knockdown affected—substrates appear prime NatA targets being affected in their steady-state levels.

This apparent discrepancy with previous studies might likely be attributed to the increased statistical power provided by the higher number of NatA substrates here found to be affected upon NatA knockdown [2,32,37]. Comprehensive shotgun analysis is however deemed necessary to extend this observation form the N-terminome to the proteome, and to further confirm this observation.

Nonetheless, in a previous attempt to study the effect of NTA on protein stability—and while for most of the substrates analyzed no such effect was discernible—we already showed that the steady-state levels of the—here also identified and confirmed—NatA substrate THOC7 (THO complex subunit 7 homolog) was strongly reduced and destabilized [32].

Despite the relative large numbers of differentially expressed proteins here inferred from their N-terminal peptide abundances and from previous differential shotgun proteomics analyses of *NAA15* haplo-insufficient and deficient iPSCs, only very few, if any, significant differences upon NatA perturbation were apparent at the mRNA level [37]. Further, and notwithstanding the significant lowered transcription and translation of *NAA15* and *NAA10* observed, we found no notable if any changes at the level of the translatome by means of ribosomal profiling (unpublished data). Thus, it can be concluded that transcriptome and translatome findings by and large rule out a possible role for NatA in steering transcription and translation efficiency apart from the NTA deficiency and NatA NTA steered protein stability effects observed.

Comprehensive NatA substrate discovery and discrimination between inefficient and inefficient substrates overall provided a better understanding of NTA by NatA. Such data is of much need to be able to further unravel the wide-ranging developmental defects observed in humans carrying mutations in *NAA10* and *NAA15* [34].

## 4. Materials and Methods

### 4.1. Cell Culture

Human A-431 cells (epidermoid carcinoma; American Type Culture Collection (ATCC, CRL-1555), Manassas, VA, USA; ATCC^®^ CRL-1555™) were cultured in 2 mM alanyl-L-glutamine dipeptide (GlutaMAX™) containing Dulbecco’s Modified Eagle Medium (DMEM) (Invitrogen, Carlsbad, CA, USA, cat n°31966047) supplemented with 10 % fetal bovine serum (HyClone, Gibco™, Waltham, MA, USA, cat n°10270106, E.U.-approved, South American origin), 100 units/mL penicillin (Gibco™, cat n°15070-063) and 100 μg/mL streptomycin (Gibco™, cat n°15070-063).

For L-Arg SILAC (stable isotope-labeling by amino acids in cell culture) labelling [40] of A-431 cells, cells were grown in DMEM SILAC medium (Invitrogen, cat n°61870-010) containing either natural (^12^C_6_) or ^13^C_6_^15^N_4_-labelled L-Arg (Cambridge Isotope Laboratories, Andover, MA, USA) at a concentration of 80 μM (i.e., 16.8 mg/L or 20% of the suggested concentration present in DMEM medium at which L-Arg to Pro conversion was not detectable for A-431 cells) and supplemented with 10% dialyzed fetal bovine serum (Invitrogen, cat n°26400-044). Cells were cultured for at least six population doublings for complete incorporation of the labelled L-Arg.

Cells were cultured at 37 °C in a humidified incubator at 37 °C and 5% CO2 and passaged every 3–4 days.

### 4.2. siRNA Transfection

siRNA mediated knockdown was performed using HiPerFect (Qiagen, Venlo, Netherlands) according to the vendors’ instructions. Gene-specific and non-targeting siRNAs (si*NAA10*, si*NAA15*) were purchased from Dharmacon (ON-TARGETplus non-targeting control siRNA pool (si*CTR*): D-001810-10, Dharmacon, GE Healthcare Life Sciences, Chicago, Illinois, USA) and siRNA targeting *NAA10* (SMARTpool siGENOME *NAA10* siRNA, Horizon Discovery, Cambridge, UK, cat n°M-009606-00-0005) and *NAA15* (ON-TARGETplus custom siRNA; 5′-GGGACCUUUCCUUACUACAdTdT-3′ (sense)) and used at the final concentration indicated, ranging between 5–100 nm to silence *NAA10* and/or *NAA15*. Cells were reverse transfected 24 h after seeding (seeding density of 2 × 10^6^ cells/10 cm plate and 2.5 × 10^5^ cells/6-well), re-transfected under identical conditions 48 h after initial transfection and harvested 96 h post initial siRNA transfection (see experimental workflow in Figure 1). In the knockdown experiment for N-terminal COFRADIC analysis, 4 × 10-cm dishes of control cells cultivated in ^13^C_6_^15^N_4_-L-Arg SILAC medium were transfected with 10 nM si-nontargeting control (si*CTR*) and 10 × 10-cm dishes of cells cultivated in ^12^C_6_ L-Arg SILAC medium were transfected with 10 nM si*NAA10*/ si*NAA15* (si*NatA*) pool. 10 μm Z-VAD-fmk pan-caspase inhibitor (R&D Systems GmbH, Wiesbaden, Germany) was added throughout the experiment (and replenished after re-transfection) to avoid induction of apoptotic cell death [39].

### 4.3. N-Terminal Proteomics

Cells were detached using cell dissociation buffer (Invitrogen, cat n°13151014). The obtained cell pellets (~9 × 10^6^ cells) were lysed in 1 mL 50 mm sodium phosphate buffer pH 7.5 and 50 mm NaCl supplemented with a complete protease inhibitor mixture tablet (Roche Diagnostics GmbH, Mannheim, Germany) using multiple cycles of freeze-thawing (3×). After lysate clearance by centrifugation for 15 min at 16,000× *g* and 4 °C, protein concentrations were adjusted to 2 mg/mL as measured using the DC Protein Assay Kit (Bio-Rad, Hercules, CA, USA)). Aliquots (50 µg) were analyzed in parallel by SDS/PAGE and Western blotting to confirm efficient knockdown of *NAA10* and *NAA15* (Figure 1).

One combined sample (Figure 1, 1/1 (si*NatA*/si*CTR*) protein mix) was subjected to N-terminal COFRADIC analysis as described in [1,44]. More specifically, N-hydroxysuccinimide ester of (stable isotopic encoded) acetate labelling of free amines at the protein level (i.e., NHS esters of ^13^C_2_D_3_ acetate) was used (i.e., in vitro ^13^C_2_D_3_-acetylation) in combination with differential L-Arg SILAC (stable isotope-labeling by amino acids in cell culture) labeling, which allows for the calculation of the extent of NTA, as well as the relative quantification of N-terminal peptides [1,52,53].

### 4.4. LC-MS/MS Analysis and Data Analysis

45 fractionated samples were introduced into the Ultimate 3000 (Dionex, Amsterdam, The Netherlands) in-line connected to an LTQ Orbitrap XL mass spectrometer (Thermo Fisher Scientific, Waltham, MA, USA) and liquid chromatography tandem mass spectrometry (LC-MS/MS) analysis was performed as described previously [44,54]. Spectra were searched against the Swiss-Prot database (taxonomy Homo sapiens) with Mascot using the Mascot Daemon interface (version 2.2.0, Matrix Science). Lys heavy (^13^C_2_D_3_^−^) acetylation (+47.036 Da), Cys carbamidomethylation (+57.021 Da) and Met oxidation (+15.995 Da) were set as fixed modifications. Variable modifications included light/heavy (^13^C_2_D_3_^−^) NTA (+42.011 Da or 47.036 Da) and pyro-Glu formation from Gln (−17.026 Da). Mass tolerance on precursor ions was set to 10 ppm (with Mascot’s C13 option set to 1) and on fragment ions to 0.5 Da. Endoproteinase semi-Arg-C/P (Arg-C specificity with Arg-Pro cleavage allowed) was set as enzyme with no missed cleavages. The peptide charge was set to 1+, 2+, 3+ and instrument setting was put to ESI-TRAP. Only N-terminal peptides that were ranked one and scored above the threshold score, set at 99% confidence, have a minimum amino acid length of six, and were compliant with the rules of NTA or initiator methionine (iMet) processing [45,46] were withheld. More specifically, iMet processing was considered in case of iMet-starting N-termini followed by any of the following amino acids; Ala, Cys, Gly, Pro, Ser, Thr, Met, or Val, or exceptionally Glu and Asp in case of actin representative of NatH type substrate N-termini [55]. Quantification of the degree of NTA was performed as described previously [1]. Considerable variations in the degree of NTA (defined as a minimum difference of 10% in the degree of NTA, or 5% in case of the clear presence of two isotopic envelopes in only one of the setups, while a unique envelope signifying an Nt-free or fully NTA N-terminus in the si*NatA* or si*CTR* setup, respectively. When additionally considering the latter category of affected N-termini, 68 N-termini (out of 182 affected N-termini in total) were further classified as being affected upon si*NatA* knockdown (Appendix A).

SILAC quantifications (^12^C_6_ L-Arg versus ^13^C_6_^15^N_4_ L-Arg) were carried out using the Mascot Distiller Quantitation Tool (version 2.2.1). Ratios for all peptides were calculated by comparing the XIC peak areas of all matched light versus heavy peptides and the calculated ratios originally reported as FALSE were all verified by visual inspection of the highest scoring MS spectra [44]. A distribution of all individual peptide ratios was determined followed robust statistics as described in [56] (Median: 1.038 and Huber scale: 0.381). Peptides displaying a ratio that was significantly up- or down-regulated (99% confidence interval) were considered affected by the *NAA10*/*NAA15* knockdown (Appendix A).

For the GO gene enrichment analysis using GOrilla a ranking of all determined N-terminal peptide-based expression ratios were used as input, keeping only the highest ranking instance of each gene. More specifically, of the 1498 unique N-termini identified in total (Appendix A), 1415 (94%) had determined values. Of these, the system recognized 1413 genes and 66 duplicate genes were removed (keeping the highest-ranking instance of each gene) leaving a total of 1347 genes. Of these, 1335 were associated with a GO term and thus used for the enrichment analysis. Only enriched (non-redundant) GO terms with minimally 5 members were considered.

Data management was done in ms_lims [57], KNIME, Excel and RStudio (R version 4.1.0). For data visualization GraphPad Prism version 9.2.0 was used.

### 4.5. SDS-PAGE and Immunoblotting

Sample loading buffer (Bio-Rad XT sample buffer) and reducing agent (Bio-Rad) was added to the samples according to the manufacturer’s instruction and equivalent amounts of protein (50 µg as measured using the DC Protein Assay Kit (Bio-Rad) and proteins were separated on a 4 to 12% on a 12% gradient XT precast Criterion gel using XT-MOPS buffer (Bio-Rad) at 150–200 V. Subsequently, proteins were transferred onto a PVDF membrane. Membranes were blocked for 30 min in a 1:1 Tris-buffered saline (TBS)/Odyssey blocking solution (cat n° 927-40003, LI-COR, Lincoln, NE, USA) and probed by Western blotting. Following overnight incubation of primary antibody in TBS-T/Odyssey blocking buffer and three 10 min washes in TBS-T (0.1% Tween-20), membranes were incubated with secondary antibody for 30 min in TBS-T/Odyssey blocking buffer followed by 3 washes in TBS-T or TBS (last washing step). The following antibodies were used: mouse anti-GAPDH (Abcam, ab9484, 1:10,000), rabbit purified IgG anti-NAA10 (anti-hARD1, Biogenes GmBH [58], 1:1000) and anti-NAA15 (anti-NATH, Biogenes GmBH [58], 1:1000), anti-mouse (IRDye 800 CW goat anti-mouse antibody IgG, LI-COR, cat n°926-32210, 1:10,000) and anti-rabbit (IRDye 800 CW goat anti-rabbit IgG, LI-COR, cat n°926-3221, 1:10,000). The bands were visualized using an Odyssey infrared imaging system (LI-COR).

## Figures and Tables

**Figure 1 ijms-22-10692-f001:**
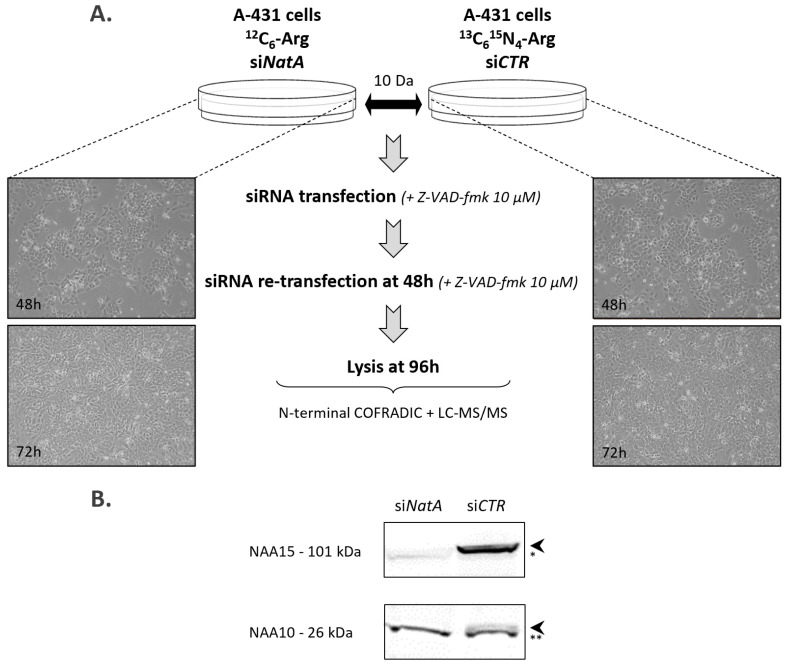
Experimental N-terminal proteomics workflow and confirmation of the efficiency of si*NatA* knockdown in human A-431 cells. (**A**) Workflow of the experimental setup used for N-terminal COFRADIC. A-431 cells cultivated in ^13^C_6_^15^N_4_ L-arginine containing SILAC medium were transfected with 10 nM non-targeting siRNA (si*CTR*) and cells cultivated in ^12^C_6_ L-arginine SILAC medium were transfected with 10 nM si*NAA10*/si*NAA15* pool (si*NatA*). Knockdown was performed in the presence of the pan-caspase inhibitor Z-VAD-fmk (10 µM). Cells were transfected 24 h after seeding, re-transfected under identical conditions 48 h after the initial transfection and harvested 96 h post initial siRNA transfection. (**B**) Immunoblots of A-431 cell lysates treated with non-targeting siRNA (si*CTR*) or si*NAA10* and si*NAA15* (si*NatA*). Blots were probed with anti-NAA10 and anti-NAA15 to assess levels of endogenous NAA10 (26 kDa) and NAA15 (101 kDa), thereby confirming highly efficient knockdown. The asterisks indicate non-specific bands with ** serving as loading control.

**Figure 2 ijms-22-10692-f002:**
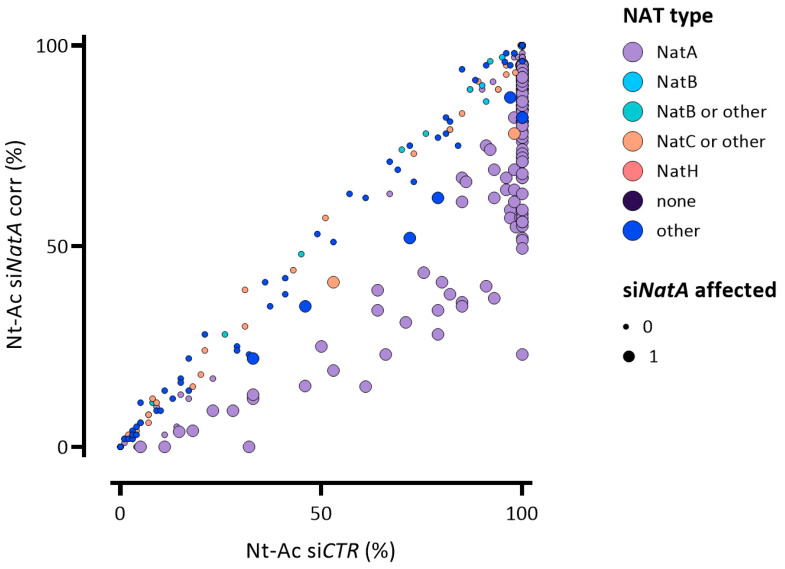
The effect of si*NatA* knockdown on the Nt-acetylome. The scatterplot displays the correlation of the determined degrees of NTA when comparing the si*CTR* (X-axis) and si*NatA* (Y-axis) N-terminome datasets (n = 1404). The enlarged dots (si*NatA* affected = 1) represent N-termini displaying a significant reduction in the degree of NTA upon si*NatA* knockdown. The circles are colour coded according to their indicated NAT substrate specificity profiles (see also NAT type column in Appendix A). si*NatA* knockdown causes a decrease in the levels of NTA as compared to the si*CTR* setup especially in case of NatA type N-termini (purple dots).

**Figure 3 ijms-22-10692-f003:**
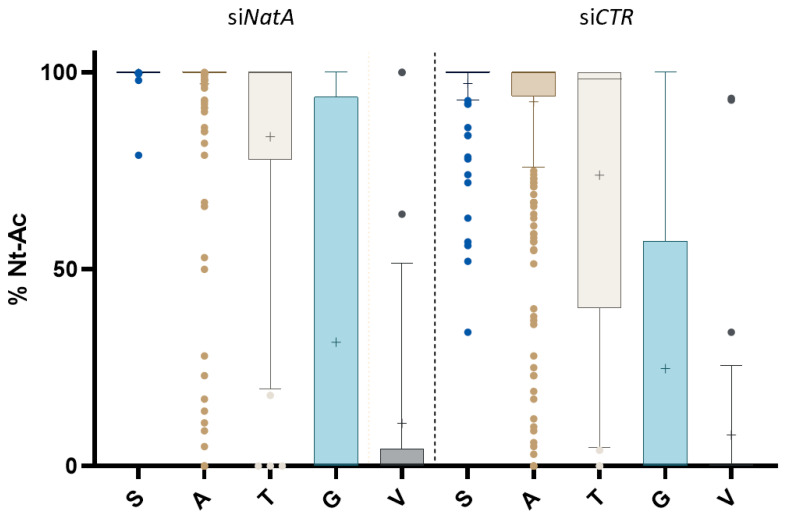
The effect of si*NatA* knockdown on the Nt-acetylation states of NatA type N-termini. The box and whiskers plot displays (differences in) the 10 to 90 percentile distribution of the degrees of NTA determined for Ser- (*n* = 172), Ala- (*n =* 458), Thr- (*n* = 40), Gly- (*n* = 38) and Val- (*n =* 33) NatA type N-termini determined in both the non-targeting siRNA (s*iCTR*) (left) or si*NAA10* and si*NAA15* (si*NatA*) (right) setups (cfr. Table 2 and Appendix A). Mean values are shown as ‘+’.

**Figure 4 ijms-22-10692-f004:**
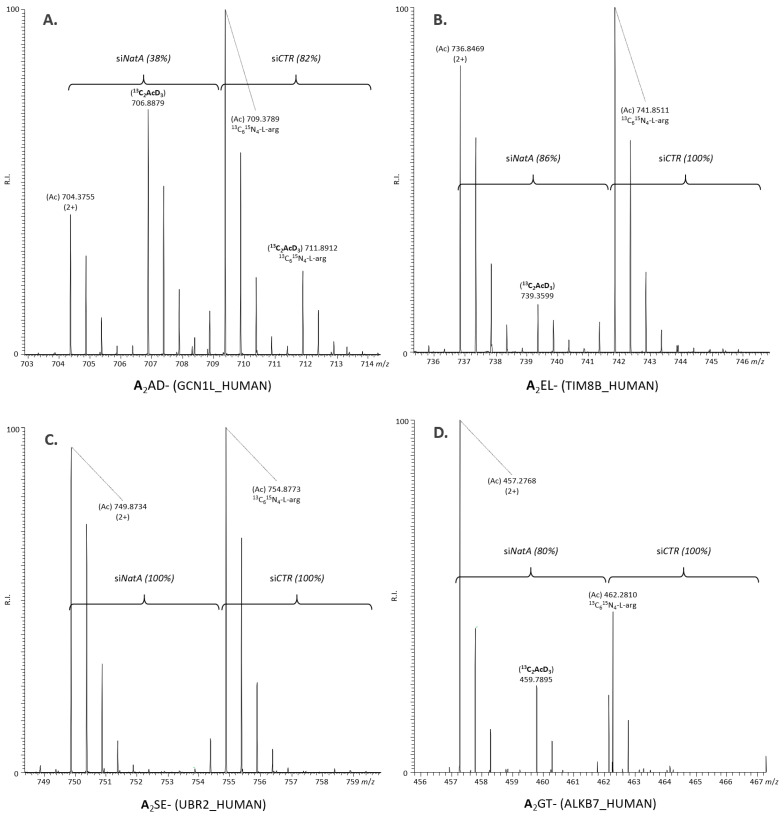
The effect of si*NatA* knockdown on representative Ala- starting NatA substrate N-termini. (**A**) An MS spectrum of the N-terminus originating from the previously confirmed NatA substrate [2,32,37], being the translational activator GCN1 (^2^AADTQVSETLKR^13^) is shown. The peptide was partially NTA in both control (si*CTR*) (82%) and si*NatA* knockdown (38%) samples. (**B**) MS spectrum of the N-terminus originating from mitochondrial import inner membrane translocase subunit Tim8 B (^2^AELGEADEAELQR^14^). The peptide was fully NTA in the control setup while being only partially NTA (86%) in the si*NatA* knockdown sample. (**C**) MS spectrum of the N-terminus originating from the E3 ubiquitin-protein ligase UBR2 (^2^ASELEPEVQAIDR^14^). The peptide was fully NTA in the control setup and was unaffected (i.e., the N-terminus remained fully NTA) upon si*NatA* knockdown. The total concentration of the representative Ala- N-termini shown in (**A**–**C**) all remained unaffected upon si*NatA* knockdown (ratios of 0.89, 0.98 and 0.93, respectively). (**D**) MS spectrum of the N-terminus originating from the mitochondrial alpha-ketoglutarate-dependent dioxygenase alkB homolog 7 (^2^AGTGLLALR^10^). The peptide was fully NTA in the control setup while being only partially NTA (80%) in the siNatA knockdown sample. The expression of this NatA substrate N-terminus was significantly increased (*p* ≤ 0.01) upon si*NatA* knockdown (ratio 2.6).

**Figure 5 ijms-22-10692-f005:**
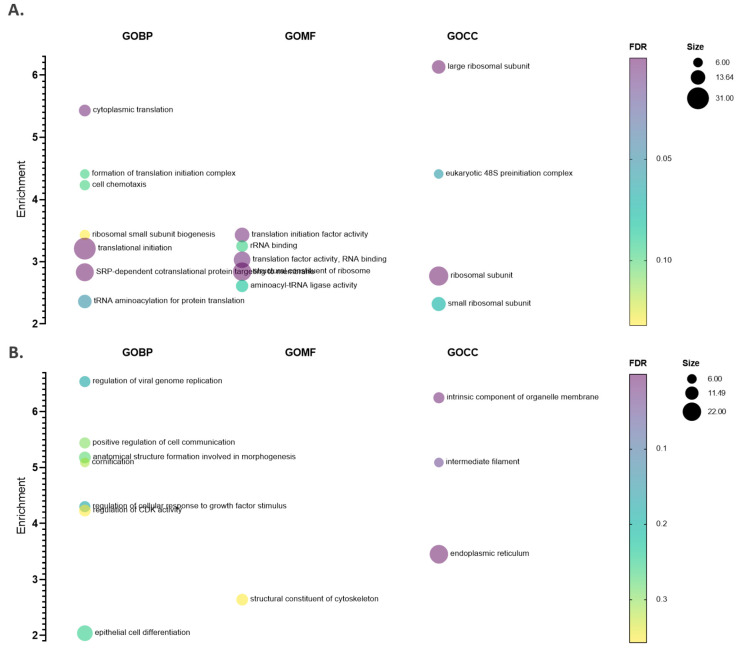
Multiple variable bubble plot representations of significantly enriched gene ontology (GO) terms. Enriched GO terms (biological process (BP), molecular function (GOMF) and cellular component (GOCC), 1335 genes) with a minimum 2-fold enrichment are shown for the si*CTR* (**A**) and si*NatA* (**B**) setup. Term enrichment was determined using GOrilla [50] and *p* values were corrected for multiple hypotheses testing using the Benjamini and Hochberg false discovery rate (FDR) (Appendix A). Only non-redundant terms with at least 5 members and corrected *p* values < 0.2 were considered. The bubble size corresponds to the number of regulated proteins with a given GO term and the colour code represents the *p*-value scale.

**Table 1 ijms-22-10692-t001:** The effect of si*NatA* knockdown on the Nt-acetylation states of NatA type N-termini. All types of N-termini identified in this study matching previously reported NatA substrate specificities ((M)A-, C-, G-, S-, T-, V-) [2,15] are listed. In all cases, the numbers of uniquely identified N-termini and the number of N-termini for which the degree of NTA could be univocally calculated/determined in both setups is indicated with their corresponding number and proportion of complete and partial NTA. Whenever a reduced degree of NTA could be observed upon si*NatA* knockdown, the corresponding number and proportion of affected N-termini are indicated. ^1^ The (X)P- rule that indicates the prevention of NTA was considered for defining putative NatA substrate N-termini (i.e., N-termini carrying a proline residue at P2’ were not considered) [48].

NatA Substrate Type (Excluding XP- N-Termini) ^1^	Identified	NTA Determined	Full + Partial NTA (%)	si*NatA* Affected (% of Total/Putative Substrates)
A-	479	458	454 (99%)	116 (25%, 26%)
C-	1	1	1 (100%)	0 (0%)
G-	43	38	14 (37%)	8 (21%, 57%)
S-	191	172	172 (100%)	26 (15%)
T-	45	40	37 (93%)	16 (40%, 43%)
V-	37	33	9 (27%)	7 (21%, 78%)
Total	796	742	687 (80%)	173 (23%,25%)

**Table 2 ijms-22-10692-t002:** Overview of the Nt-acetylation status of all identified protein N-termini. N-termini are categorized according to their NAT substrate class (with totals of substrate classes italicized) and the setup analyzed. Notable differences in NTA levels between the si*CTR* and si*NatA* setups were enriched in the NatA category of N-termini (See also Table 1 and Appendix A for the complete list of all (affected) N-termini identified). Only N-termini of which the degree of NTA could be determined and which were compliant with the rules NTA and NME [46,49] were used for the overall calculation of NTA.

			Setup							
			si*NatA*				s*iCTR*			
NAT Type	M-Starting	P1’ (P2’)	Free	Partial NTA	Fully NTA	Total	Free	Partial NTA	Fully NTA	Total
NatA	0	A	4	134	320	458	4	25	429	458
		C			1	1			1	1
		G	26	6	6	38	24	6	8	38
		S		30	142	172		1	171	172
		T	3	17	20	40	3	9	28	40
		V	26	7	0	33	24	7	2	33
	*Total*		*59*	*194*	*489*	*742*	*55*	*48*	*639*	*742*
Other (iMet-retained NatA-type N-termini)	1	A		2	10	12		2	10	12
		G	2	2	2	6	2	2	2	6
		S	1	1	3	5	1	1	3	5
		T	1	10	16	27	1	8	18	27
		V	2	12	11	25	2	12	11	25
	*Total*		*6*	*27*	*42*	*75*	*6*	*25*	*44*	*75*
NatB	1	D		1	101	102		1	101	102
		E		3	186	189		3	186	189
		N		3	44	47		3	44	47
	*Total*			*7*	*331*	*338*		*7*	*331*	*338*
NatB or other	1	Q	1	4	14	19		5	14	19
	*Total*		*1*	*4*	*14*	*19*	*0*	*5*	*14*	*19*
NatC or other	1	F	2	2	14	18	3	1	14	18
		I	2	6	5	13	2	6	5	13
		L	4	15	25	44	3	14	27	44
		Y	1	3	6	10	1	3	6	10
	*Total*		*9*	*26*	*50*	*85*	*9*	*24*	*52*	*85*
NatF or other	1	K	19	20	8	47	18	21	8	47
		M		1	12	13		1	12	13
	*Total*		*19*	*21*	*20*	*60*	*18*	*22*	*20*	*60*
None	0	P	62			62	62			62
	0	X(P)	15			15	15			15
	1	P	4			4	4			4
	*Total*		*81*	*0*	*0*	*81*	*81*	*0*	*0*	*81*
NatH	1	D			1	1			1	1
	*Total*		*0*	*0*	*1*	*1*	*0*	*0*	*1*	*1*
Grand Total			175	279	947	1401	169	131	1101	1401

## Data Availability

N-terminal proteomics data is contained within ‘Appendix A’ of the article. The corresponding raw MS-data are made available under the Open Science Framework projects ‘nw32r’ (https://osf.io/nw32r/, accessed on 25 August 2021) and ‘c9kyx’ (https://osf.io/c9kyx/, accessed on 25 August 2021).

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
