# Peer review of "Charting the N-Terminal Acetylome: A Comprehensive Map of Human NatA Substrates"

_ijms, 2021, doi:10.3390/ijms221910692_

Round 1

Reviewer 1 Report

The author performed COFRADIC N-terminomics on a cell and the same cell with NatA knocked down. The N-termini proteomics seems to be well performed and summarized in several supplementary files. However, it is ambiguous whether the significance of the data analysis has been sufficiently explained. For example, 60 proteins had n-termini at both positions 1 and 2, how was the result of these data processed? How was NTA% calculated when there was a quantitative change in protein expression level (SILAC N-termini level)? According to the description on the result, 10% and 5% changes was considered significant, but how many experiments were repeated? What is the rationale for such a change to be significant?

The author claimed 99% knocked-down by Western blot. Is this similar efficiency seen in qRTPCR or the SILAC result? More importantly, is the 1% remaining NatA proteins enough for acetylation of 75% NatA substrates showing no change? The author explains so. If then, it is likely that NatA's substrates are already known and the author just showed how many of them could be affected at the level of the knock down he attained, but then I don't know what the charting means. Without other types of validation or comparative analysis of NTA at various conditions, I do not konw what more the charting of NatA substrates can give to the NTA field. 

Author Response

For the responses to the reviewers comments, please see the attachment.

Reviewer 2 Report

Petra Van Damme investigated the N-terminal acetylome by performing SILAC-N-terminus proteomics in human A-431 cells -/+ knockdown of NAA10 and NAA15, two subunits of the N-alpha acetyltransferase, by siRNAs. Following knockdown, the author observed a higher number of N-terminus with a decreased acetylation compared to previously published reports. In addition, a small number of proteins were found to have their stability affected by the knockdown of the NAA10 and NAA15.

Overall, the manuscript expands the number of proteins found experimentally to be regulated by the N-alpha acetyltransferase complex.

I have a few comments:

  1. How many biological replicates have been performed for the proteomics? This should be indicated in the Methods.
  2. Figure 5: How many genes have been used for the GO enrichment analysis? Is it based on the proteins found in Supplementary Table 2 or 3?

Author Response

(The authors gave the same response as above.)

Round 2

Reviewer 1 Report

I read the response letter carefully. The author explained well how to calculate NTA% . However, most of those explanations, especially about the reproducibility are about his/her past papers, not the current data. The results presented in this manuscript are data that were performed only once with control and knowckdown as a pair.

But more than that, I was asking what were new findings as a result of the studies. According to the author's explanation, NatA's substrates are already known. How many of what are known as substrates turned out to be not actually substrates, or vice versa? It is very hard to agree that this study changed the existing NatA chart, that is, charted the NatA substrates. I see that only how much% of the chart is covered by the author's studies has changed.